# Mapping and Monitoring of Land Cover/Land Use (LCLU) Changes in the Crozon Peninsula (Brittany, France) from 2007 to 2018 by Machine Learning Algorithms (Support Vector Machine, Random Forest, and Convolutional Neural Network) and by Post-classification Comparison (PCC)

**Guanyao Xie [1,*] and Simona Niculescu [1,2]**

1   Laboratory LETG-Brest, Géomer, UMR 6554 CNRS, IUEM UBO, 29200 Brest, France;
    Simona.Niculescu@univ-brest.fr
2   Department of Geography, University of Western Brittany, 3 Rue des Archives, 29238 Brest, France
*   Correspondence: Guanyao.Xie@univ-brest.fr

**Abstract:** Land cover/land use (LCLU) is currently a very important topic, especially for coastal areas that connect the land and the coast and tend to change frequently. LCLU plays a crucial role in land and territory planning and management tasks. This study aims to complement information on the types and rates of LCLU multiannual changes with the distributions, rates, and consequences of these changes in the Crozon Peninsula, a highly fragmented coastal area. To evaluate the multiannual change detection (CD) capabilities using high-resolution (HR) satellite imagery, we implemented three remote sensing algorithms: a support vector machine (SVM), a random forest (RF) combined with geographic object-based image analysis techniques (GEOBIA), and a convolutional neural network (CNN), with SPOT 5 and Sentinel 2 data from 2007 and 2018. Accurate and timely CD is the most important aspect of this process. Although all algorithms were indicated as efficient in our study, with accuracy indices between 70% and 90%, the CNN had significantly higher accuracy than the SVM and RF, up to 90%. The inclusion of the CNN significantly improved the classification performance (5–10% increase in the overall accuracy) compared with the SVM and RF classifiers applied in our study. The CNN eliminated some of the confusion that characterizes a coastal area. Through the study of CD results by post-classification comparison (PCC), multiple changes in LCLU could be observed between 2007 and 2018: both the cultivated and non-vegetated areas increased, accompanied by high deforestation, which could be explained by the high rate of urbanization in the peninsula.

**Keywords:** remote sensing; machine learning; GEOBIA; CNN; land cover/land use; SPOT 5; Sentinel 2; change detection

## 1. Introduction

Coastal zones are the shores of seas or oceans. Today, nearly half of the world's population lives in coastal regions where multiple activities are developed [1]. Over the last century, coastal zones throughout the world have undergone major changes related to a significant influx of the population. Coveted, densely populated, and exploited by human societies, coastal zones are therefore subject to significant pressures that generate territorial dynamics and changes in land cover/land use (LCLU). LCLU is al-ways influenced by human actions and environmental features and processes, and it mediates the interactions of these two factors. This means that land use changes are primarily due to human actions, which are associated with economic development, tech-neology, environmental change, and especially, population growth, which usually has parallel rates to land use change [2,3]. However, traditional methods require direct observations in the field; usually, they are not

only ineffective, expensive, time-consuming, and labor intensive but are also limited on the local scale. Hence, remote sensing with analysis techniques is highly recommended, and there has been an in-creasing demand for LCLU studies since the first launch of Earth observation satellites in 1972 with Landsat-1. Since that time, the monitoring and mapping of LCLUs over large areas and in a consistent manner has been made possible with Earth observation (EO) data, and detection of these changes by EO data is necessary for the better management of territory and resources. Moreover, each new generation of satellite equipment increases the resolution of sensors that collect high spatial resolution data for LCLU mapping and monitoring [4]. Since high-resolution satellite images are now available, land cover change mapping and monitoring at the landscape or local scale have been developed at a high rate of speed [5–7].

Several national and international organizations have produced regional land change maps that represent a location on a single date (e.g., CORINE Land Cover 2000 in Europe), with Landsat observations acquired in a target year interval (e.g., ±1–3 years). Some programs repeat land cover mapping periodically (e.g., NLCD 2001/2006/2011 in the United States and CSBIO in France) to allow the observation of the changes. The local accuracy of these global or national land cover maps generated from coarse spatial resolution data is low, especially in regions with fragmented land covers [8].

At the same time, for studies at larger scales, satellite data have been used to monitor LCLU changes worldwide in various fields of research, such as mapping cropland conversions [9], monitoring urbanization and its impacts [10–12], monitoring deforestation [13–17], evaluating the environment [18–20] and biodiversity losses, and examining the influence of LCLU on climate change [21]. Nonetheless, all types of land use might lead to detrimental impacts and effects in many fields: for example, the abandonment of agricultural land without restoration is linked to a specific set of problems, including landscape degradation and an increased risk of erosion [4]. These irreversible impacts of LCLU change have significantly increased in recent decades, and so the mapping and monitoring LCLU is very important as the first step in the study and management of this phenomenon.

In recent years, given the importance of LCLU changes and the increasing availability of open-access archived multitemporal datasets, many methods for analyzing and mapping LCLU changes have been developed. The diversity of algorithms for studying LCLU changes was also determined by the diversity of remote sensing sensor types (e.g., multispectral, hyperspectral, and SAR). Among the most commonly used satellite images in change detection (CD) studies are multispectral images due to the diversity of the types of sensors used to collect the data and the high temporal resolution of datasets for this type of study. For example, Wang et al. 2018 [22] conducted a study in a coastal area of Dongguan City, China, using SPOT-5 images acquired in 2005 and 2010. In this study, a scale self-adapting segmentation (SSAS) approach based on the exponential sampling of a scale parameter and the location of the local maximum of a weighted local variance was proposed to determine the scale selection problem when segmenting images constrained by LCLU for detecting changes. Tran et al. 2015 [23] conducted a study in coastal areas of the Mekong Delta on changes in LCLU between 1973 and 2011 from Landsat and SPOT images. The supervised maximum likelihood classification algorithm was demonstrated to provide the best results from remotely sensed data when each class had a Gaussian distribution. Guan et al. 2020 [24] studied a CD and classification algorithm for urban expansion processes in Tianjin (a coastal city in China) based on a Landsat time series from 1985 to 2018. They applied the c-factor approach with the Ross Thick-LiSparse-R model to correct the bi-directional reflectance distribution function (BRDF) effect for each Landsat image and calculated a spatial line density feature for improving the CD and the classification. Dou and Chen 2017 [25] proposed a study in Shenzhen, a coastal city in China, from Landsat images using C4.5-based AdaBoost, and a hierarchical classification method was developed to extract specific classes with high accuracy by combining a specific number of

base-classifier decisions. According to this study, the landscape of Shenzhen city has been profoundly changed by prominent urban expansion.

In addition, in recent decades, remote sensing techniques have progressed, and many methods, such as machine learning, have been developed for LCLU change studies, such as support vector machines (SVMs), random forests (RFs), and convolutional neural networks (CNNs). Nonparametric machine learning algorithms such as SVM and RF are well-known for their optimal classification accuracies in land cover classification applications [26–28]. These algorithms have significant advantages and similar abilities in classifying multitemporal and multisensor data, including high-dimensional datasets and improved overall accuracy [29,30]. The accurate and timely detection of changes is the most important aspect of this process. Moreover, CNN, a more recently developed but well-represented deep learning method, allows the rapid and effective analysis and classification of LCLUs and has proven a suitable and reliable method for accurate CD in complex scenes. Although it is more recent, many studies have made use of this method. Wang et al., 2020 [31] proposed a new coarse-to-fine deep learning-based land-use CD method. In this study, several models of CNN well-trained with a new scene classification dataset were able to provide ac-curate pixel-level range CD results with a high detection accuracy and reflect the changes in LCLU in detail. In another study of Han et al., 2020 [32], a weighted Demptster-Shafer theory fusion method was proposed. This method achieved reliable CD results with high accuracy using only two very high-resolution multitemporal images by generating object-based CD through combining multiple pixel-based CDs.

At the same time, in the Pays de Brest, which the Crozon Peninsula is part of, a category of LCLU has been studied through shallow machine learning algorithms. Niculescu et al. 2018 [33] and Niculescu et al. 2020 [34] applied the algorithms of rotation forest, canonical correlation forests and random forest (RF) with satisfactory results for the classification of the different categories of land cover (vegetation) of the peninsula, as well as the summer and winter crops from the synergy of optical and radar data from the Sentinel satellite.

LCLU changes in coastal areas have been studied with machine learning algorithms in different environments. Munoz et al 2021 [35] analyzed the coastal wetland dynamics associated with urbanization, the sea level rise and hurricane impacts in the Mobile Bay watershed since 1984. They developed a land cover classification model with CNNs and a data fusion (DF) framework. The classification model achieved the highest overall accuracy (0.93) and f1-scores in the woody (0.90) and emergent wetland classes (0.99) when those datasets were fused into the framework.

More methodological work on the application of CNNs for CD was conducted by Jing et al. 2020 [36]. In this study, a CD method was proposed that combines a multiscale simple linear iterative clustering-convolutional neural network (SLIC-CNN) with stacked convolutional auto encoder (SCAE) features to improve the CD capabilities with HR satellite images. This method uses the self-learning SCAE architecture as the feature extractor to integrate multiscale, spectral, geometric, textural and deep structural features to enhance the characteristics of ground objects in images.

Machine learning methods were combined with Object-based Image Analysis (OBIA) techniques by Jozdani et al., 2019 [37] for urban LCLU classification. The multi-layer perceptron model led to the most accurate classification results in this study. However, it is also important to note that GB/XGB and SVM produced highly accurate classification results, demonstrating the versatility of these ML algorithms.

In this work, we aimed to study multiannual changes of LCLU in the Crozon Peninsula, an area that has mainly been marked by conversion between three types of LCLU: cropland, urban, and vegetation, in recent years, especially from 2007 to 2018. The challenge of this research was to deal with multiannual changes of a coastal area with different shapes and patterns by combining machine learning methods with PCC. To improve the CD capabilities using high-resolution satellite images, we implemented three remote sensing machine learning algorithms: SVM, RF combined with GEOBIA techniques, and CNN with SPOT 5 and Sentinel 2 data from 2007 and 2018, all effective and valid data sources. An

evaluation of these three advanced machine-learning algorithms for image classification in terms of the overall accuracy (OA), producer's accuracy (PA), user's accuracy (UA), and confidence interval was conducted to more precisely detect the type of multiannual change.

## 2. Study Area

The study area, the Crozon Peninsula canton south of the Landerneau-Daoulas canton, is located on the west coast of France in the Pays de Brest, Department of Finistère and the region of Brittany (Figure 1).

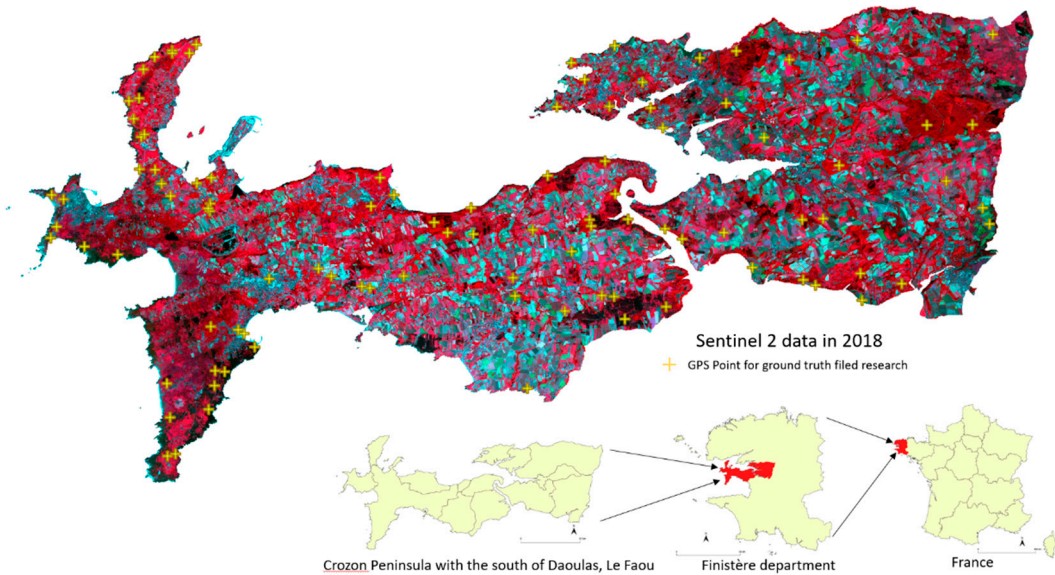

**Figure 1.** Location of the study area, including the Crozon Peninsula and two bordering regions, located in Pays de Brest, Finistère, Brittany, France, with the RGB band combination for Sentinel 2 (2018) and the location of the ground truth field research.

It covers a land area of 365.4 km$^2$ that extends between latitudes 48° 10′04″ N and 48° 21′28″ N, longitudes 4°02′44″ W and 4°38′37″ W. The Crozon Peninsula is a sedimentary site with contrasting topography and contours that separate the Bay of Brest and the Bay of Douarnenez. The region is a mosaic of cliffs, dunes, moors, peat bogs, and coastal wetlands. The peninsula thus presents phytocenetic, faunistic, and landscape interests. The population of the study area is 29,893; this makes the population density approximately 81.6 per km$^2$. The topography of the Crozon Peninsula is mostly dominated by plains, except for hills in the east and northeast, and the elevation of the area ranges between 0 m and 300 m. Climatically, the study area is classified as type Cfb (temperate oceanic climate) according to the Köppen climate classification. On average, the Crozon Peninsula reaches 1208 mm of precipitation per year, and the annual average temperature is 12.2 °C. The land cover is characterized by forest, shrubs, and grasslands, which are mostly in the west, urban areas, cropland (including mainly corn and wheat) and meadow.

Traditionally, the majority of local people practice agricultural or related activities in the Finistère Department, in which 57% of the department's surface is devoted to agricultural use. However, the French National Defense provides more than half of the employment in the Crozon Peninsula; hence, other activity sectors (e.g., agriculture, industry, construction and commerce) are proportionally less important.

Nevertheless, the land cover was actually in sharp transition in our study area between 2008 and 2018, with the peninsula especially marked by an increasing service and commerce sectors. Therefore, the study area was chosen as a typical ideal case to study land cover changes.

## 3. Data

Operable high-quality cloud-free satellite images in this area are extremely rare due to the annual high-intensity rainfall and, hence, heavy cloud cover. Despite these limitations, three cloud-free images from two dates in 2007 and 2018 with the same scene area were acquired from either the SPOT or Sentinel platforms to study land cover changes in the study area during the summer, which is the growing season for crops (Table 1).

**Table 1.** Satellite images used in the study.

| Date | Satellite | Spectral Bands | Spatial Resolution |
|---|---|---|---|
| 2 July 2007 | SPOT-5 | Green, Red, Near-infrared | 10 m resampled to 2.5 m |
| 24 June 2018 | Sentinel-2B | 1, 2 (Blue), 3 (Green), 4 (Red), 5, 6, 7, 8 (Near-infrared), 8A, 9, 10, 11, 12 | 10 m |
| 24 June 2018 | Sentinel-2B | 1, 2 (Blue), 3 (Green), 4 (Red), 5, 6, 7, 8 (Near-infrared), 8A, 9, 10, 11, 12 | 10 m |

First, a SPOT-5 satellite image was obtained from the early summer of 2007. SPOT-5 was the fifth satellite in the SPOT series of CNES (Space Agency of France). It was launched in 2002 and completed its mission by the end of 2012. It provided very high spatial resolution (2.5 m in the panchromatic band and 10 m in the multispectral band) and wide-area coverage satellite images with a revisit frequency of 2 to 3 days [38]. The multispectral SPOT-5 image downloaded from the ESA was obtained by merging the 2.5 m panchromatic band and the 10 m multispectral band, resulting in the spatial information of the image being identical to the information observed with the panchromatic sensor (earth.eas.int).

Second, Sentinel-2 is an imaging mission that operates in the frame of the Copernicus (ex-GMES Global Monitoring for Environment and Security) program, which is implemented by the European Commission (EC) and the European Space Agency (ESA). The twin Sentinel-2 satellites (2A and 2B) deliver continually polar-orbiting; multispectral; high-resolution (10 m spatial resolution for B2, B3, B4, and B8; 20 m for B5, B6, B7, B8a, B11, and B12; and 60 m for B1, B9, and B10); high revisit frequency (10 days of revisit frequency for each satellite and a combined revisit frequency of five days); wide-swath; and open-access satellite imagery [39]. Two level 2A atmospheric effect-corrected Sentinel-2 images of the same date in the middle of the summer in 2018 were acquired from Theia (catalog.theia-land.fr); a mosaic was then created by combining two images to cover the whole study area, and four spectral bands at a 10 m resolution (red, green, blue, and near-infrared) were extracted for further use.

For the purpose of land cover identification at the sample selection step, we also used Google Earth and RPG (Graphic parcel register) maps and a French database with agricultural parcel identification as the reference data, complemented by observation and survey in the field when necessary.

## 4. Methods

The methodology of this paper is detailed in three main parts as follows: preprocessing, image processing, and postprocessing. Three satellite images of two dates were processed in QGIS (SAGA, Grass, OrfeoToolbox7.3.0), eCognition 9.5 and 10.0. A flow chart of the proposed global methodology and details of the CNN are displayed below (Figures 2 and 3).

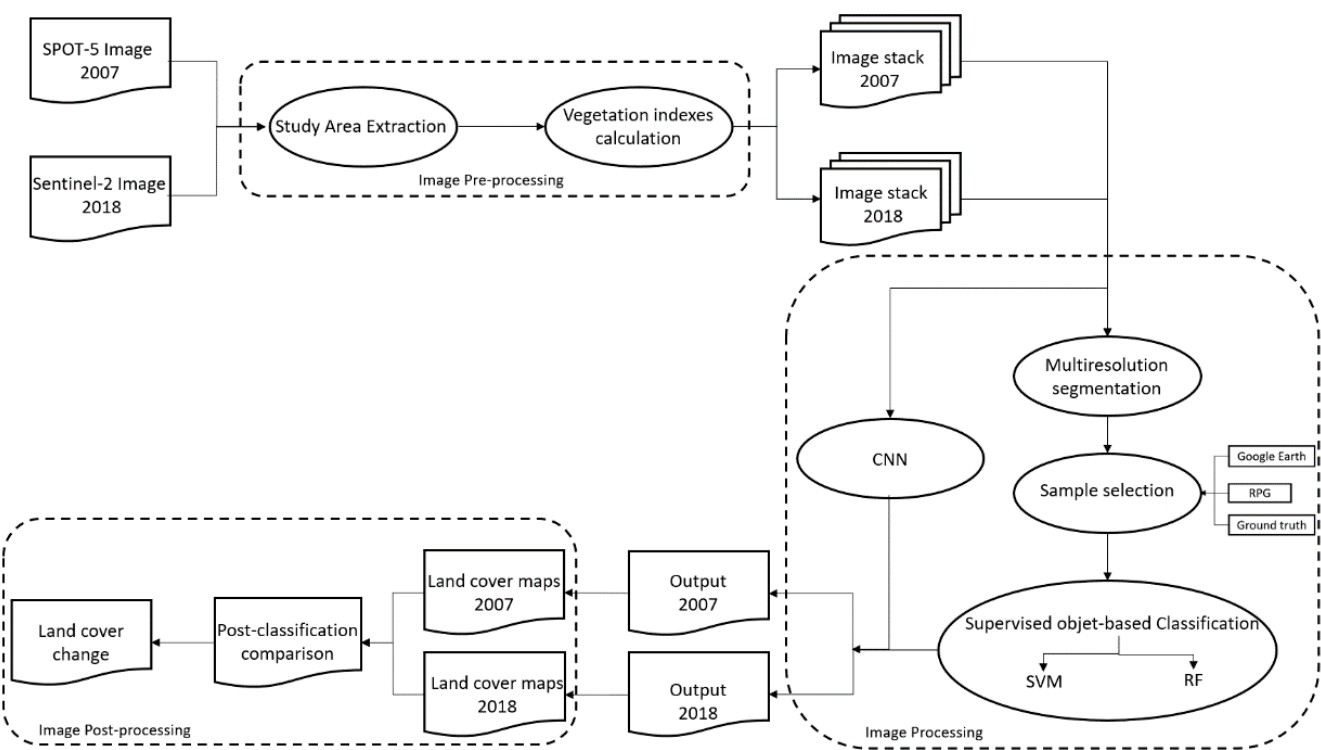

**Figure 2.** Global methodology proposed.

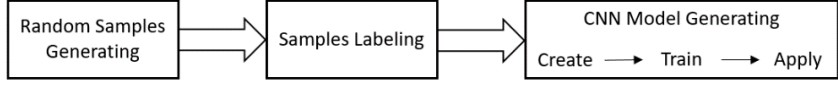

**Figure 3.** Detailed CNN methodology.

*4.1. Image Preprocessing*

4.1.1. Study Area Extraction

After satellite image acquisition, a mosaic of the two images of the same date in 2018 was created to cover the whole study area. Then, the boundary of the Crozon Peninsula and south of Landerneau-Daoulas were used to extract our area of interest by applying subsets to raw images to reduce the image size, processing time and storage space.

4.1.2. Vegetation Indices Calculation

The vegetation index is a qualitative and quantitative evaluation of vegetative cover and growth dynamics using spectral band measurements, which have been proven to have better sensitivity than individual spectral bands in identifying vegetated areas or different vegetation types and evaluating the vegetative cover density [40,41].

Due to the different spectral bands used and their ratios, the results are also different, not only because the reflectance of vegetation to the electromagnetic spectrum is determined by the chemical and morphological characteristics of the surfaces of the organs or leaves of the plants [42,43] but also because the values are heavily influenced by the atmosphere, sensor calibration, sensor viewing conditions, soil moisture, soil color, and brightness [41].

For this reason, different indices highlight different specific properties of vegetation features, and thus, more than 100 vegetation indices have been developed by scientists for various purposes and specific applications. Three of these were utilized in our study.

- The normalized difference vegetation index (NDVI), the most known and widely used vegetation index, was proposed in 1973 by Rousse et al. [44]. This index is a normalized ratio between the red and near-infrared spectral bands, as follows:

$$NDVI = \frac{NIR - RED}{NIR + RED} \tag{1}$$

Although the NDVI is widely used in research related to regional or global vegetation monitoring, some limitations remain, such as a sensitivity to the effects of soil brightness, soil color and a series of atmospheric effects [43].

- The green normalized difference vegetation index (GNDVI), the index proposed in 1996 by Gitelson et al. [45], is very similar to the NDVI; nonetheless, it considers the green spectral band an instance of red, and the expression is as follows:

$$GNDVI = \frac{NIR - Green}{NIR + Green} \tag{2}$$

The GNDVI is proven to be more sensitive to chlorophyll than the original "red" band and enabled a precise estimation of the pigment concentration [45].

- The Enhanced Vegetation Index 2 (EVI2), the two-band index, was developed by Jiang et al. in 2007 [46] as an adaptation of the enhanced vegetation index (EVI) that was designed to improve on its sensitivity in high biomass regions while minimizing the soil background signals and atmospheric influences. However, since the role of the blue band in the EVI only reduces noise, the EVI2 was developed without the blue band to maintain the soil-adjustment and linearization functions in the EVI but to break through its limit to sensor systems [46]. The index is expressed as follows:

$$EVI2 = 2.5 \frac{N - R}{N + 2.4R + 1} \tag{3}$$

After calculating the three vegetation indices, we created an image stack with the original spectral bands and all of the indices for image processing.

### 4.2. Image Processing

4.2.1. Shallow Machine Learning Methods (SVM and RF)

In this study, supervised object-based classification was performed on two image stacks of two different years. Segmentation was applied first, followed by two nonparametric machine learning algorithms. SVM and RF were trained and applied in this step.

Multiresolution Segmentation (MRS)

Segmentation is the first processing step of object-oriented image analysis. MRS is one of the most successful region-based segmentation algorithms [47] and is based on homogeneity by extracting meaningful image objects with a reasonable processing speed. At the same time, the texture, color, form, spectra, and sizes of objects are accounted for [48]. The process starts by considering each pixel as an individual object; afterwards, pairs of adjacent image objects are merged to form larger segments [49]. The scale, compactness, and shape are the main parameters of the merging decision of the algorithm. Among the three parameters, the scale parameter allows users to define the maximum standard deviation of the heterogeneity used for image segmentation controlling the amount of spectral variation within objects and the size of their results [47,50]. There are two compositions of homogeneity criteria, which are the weight of the shape criterion and the compactness criterion [51]. The shape parameter is a weighting between the shape and the spectral information of the objects. When the parameter is 0, only color is considered. Then, the higher the value, the more important the shape is. The compactness parameter defines the weight of the compactness criterion, which represents the compactness of the objects formed during segmentation. The higher the value, the more compact objects are [51].

In this study, the scale, compactness, and shape parameters used were assigned as follows: 10, 0.1, and 0.3, respectively. The selection of the parameters was completed on a trial-and-error basis.

Sample Selection

Supervised methods were performed in our research, of which the goal was to build a concise model of the distribution of class labels in terms of the predictor features [52]. In contrast to the unsupervised methods, users are able to provide knowledge and experience to the process with these methods. Sample selection is an indispensable step in training machine-learning models using supervised methods. In this study, all of the samples presented in Table 2 were selected manually with Google Earth, an RPG (Graphic parcel register) map, and ground truth as the reference data, and the ground truth values were taken during a field survey with a Global Positioning System (GPS) in August. The samples were then used to train two classifiers in the next step.

**Table 2.** Training samples surface area for SVM and RF model training in 2007(2a) and 2018(2b).

| Class | Area surface of training samples (km$^2$) |
|---|---|
| Cropland | 13.68 |
| Cropland with bare soils | 18.18 |
| Water | 00.39 |
| Vegetation | 35.61 |
| Non-vegetation | 03.86 |
| Total | 71.72 |
| 2a | |
| **Class** | **Area surface of training samples (km$^2$)** |
| Cropland | 11.76 |
| Cropland with bare soils | 17.75 |
| Water | 00.14 |
| Vegetation | 41.08 |
| Non-vegetation | 07.69 |
| Total | 78.42 |
| 2b | |

SVM Classification

SVM, also called the Support Vector Network, is one of the most robust and frequently used supervised nonparametric statistical machine-learning methods. It is capable of generating good classification results with a simple training dataset in comparison to many supervised learning methods [53]. Originally, SVM was a learning machine with the aim of solving a binary classification problem [54]. SVM is capable of handling two different cases: when the classes are linearly separable, the machine seeks a linear decision boundary called a hyperplane that minimizes the generalization error and leaves the greatest margin between the two classes [55]. In contrast, in the case of nonlinearly separable classes, a method of projecting the input data onto a high-dimensional feature space with kernel functions was proposed, which worked in such a way that the problem is transformed into a linear classification problem in that space [54,55].

Some multiclass classification methods were developed for cases in which this initially two-group classification learning machine faces a multiclass problem. The most commonly used strategies are described as "one against one" and "one against all." Traditionally, SVM has always been considered to be a pixel-based classification method, and it always obtains great classification results in this way [56–60]. However, some studies have proven that SVMs can also produce very satisfactory results as object-based classifiers [61–63], which involves spectra, texture, form and shape information [64]. Therefore, the SVM is tested and evaluated as an object-based classifier in this paper. In this method, segmentation was previously completed [64].

The training and classification of the SVM module are applied using ECognition software with a radial basis function (RBF) kernel. The SVM kernel is a set of mathematic functions for taking sequence data as the input and then transforming them into the required form of processing data. This function can transform a non-linear problem into a

linear equation in a higher-dimensional feature space. RBF, a very effective and accurate kernel type, is capable of performing the transformation with the radial basis method in the case of lack of prior knowledge about the data [65,66]. Furthermore, the module was executed with 10 as its capacity constant, also called the c-parameter, with the aim of minimizing error function and avoiding a misclassification problems. The higher the c-parameter value, the smaller-margin hyperplane the optimizer looks for [51].

RF Classification

Other than the SVM and most classifiers, RF is a combination of multiple tree-based classifiers to produce a single classification, an ensemble of decision trees, where each single tree contributes a vote for the assignment of the most popular class to the input data [55,67–69].

This type of ensemble method has been highly developed and used for two decades and has been proven to make significant progress in the classification accuracy for land cover classification. In particular, RF can address thousands of input data without variable deletion and estimates the importance of the variables in the classification [55,68,70–72].

The RF classification requires two important user-defined parameters to train the model: Two parameters are set on a trial-and-error basis: the number of decision trees grown in the forest; decision trees are capable of contributing a prediction then voting for the final model's prediction; this parameter was set to 300 and 200 for 2007 and 2018, respectively. In addition, the maximum tree depth, which means the length of each tree in the forest; generally, a larger tree can capture more information about the data with the more splits it has; this parameter was defined as 20 for both years.

4.2.2. Deep Learning Method (CNN)

As a subset of machine learning, the CNN was inspired by the functioning of the nervous system of the human brain; it utilizes artificial neural networks (ANNs) but has multiple layers. CNNs are mainly designed for image classification [73,74]. They are well-suited for solving complex problems and recognizing image objects with revolutionary accuracy levels that none of the other machine learning approaches have yet achieved [51]. The CNN implemented in eCognition is based on the Google TensorFlow library.

It has an input layer that consists of an image patch, at least one hidden layer, and an output layer where the classified output has a unit for each class that the network predicts. Images as input layers must go through multiple hidden layers for the output to be obtained (Figure 4).

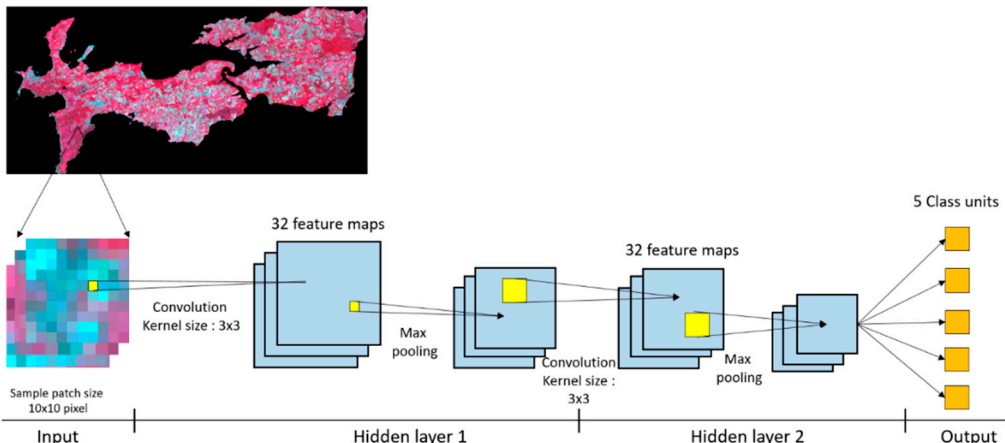

**Figure 4.** CNN model proposed and used for the classification of images from 2018 in the study.

Generating Labeled Samples

The training of a CNN model requires a large number of samples because with a larger training sample, a better model is trained, and a higher accuracy level can be achieved [73,74]. All of the training data in this study were prepared manually to obtain better accuracy. For years between 1960 and 1980, the samples in the form of points were automatically and evenly selected in QGIS by random selection for the purpose of avoiding bias. We then categorized all of the points manually into five distinct classes and created sample patches by including all of the pixels that surround each point for the model training. The algorithm then shuffled the labeled sample patches and created a random sample order for training [51].

Model Generation

Two models were created separately for 2007 and 2018. Each model had two hidden layers, a kernel size for the convolution, which is a matrix used to extract the features from the image, a number of feature maps, defined as the output activations for one filter applied to the previous layer, and a max-pooling step that can significantly reduce the number of units by keeping only the maximum response of several units from the first stage [51]. For the processing of the 2007 images, the batch size, the number of training examples utilized in one iteration, was set to 32, the kernel size was assigned to be $3 \times 3$ with 64 feature maps in the first hidden layer, and the second hidden layer had a kernel size and feature map of $4 \times 4$ and 64, respectively. For the 2018 images, the batch size was set to 10, and both hidden layers were assigned $3 \times 3$ and 32 for the kernel size and number of feature maps, respectively. Both hidden layers of the two models contained a max-pooling stage using a $2 \times 2$ filter. Then the two models were trained based on the trial-and-error method, with a learning rate of 0.001. After obtaining a satisfactory CNN accuracy, the two models were validated and used to produce the classification of two satellite images, from 2007 and 2018 separately.

### *4.3. Image Postprocessing*
### 4.3.1. Accuracy Assessment

The accuracy assessment, a principal component of land cover classification, is used to express the classification's degree of agreement with reality [75–77]. The accuracy assessment statistics of the classifiers (SVM, RF and CNN), based on confidence intervals [78], were calculated for each method and each class to check the model training and classification quality by comparing the classification with the reference values. The accuracy assessment used in this study included three indices: the overall accuracy (OA), the producer's accuracy (PA) and the user's accuracy (UA) —which are among the best-known and most highly promoted quantitative accuracy assessment metrics for the evaluation of classification quality or for comparisons among different classifications.

The OA is the probability that something will be correctly classified by a classifier. It is computed by dividing the total number of correct pixels by the total number of pixels in the error matrix [76–79]. The PA is a measure of errors of omission; it refers to instances in which something is erroneously excluded from consideration when it should have been included. On the other hand, the UA measures the error of commission, which refers to something that is erroneously included for consideration when it should have been excluded [80–82].

The indices of the accuracy assessment were generated with an algorithm from Olofsson based on the confidence interval. Therefore, all indicators presented in the tables are followed by an uncertainty rate. A higher uncertainty signifies that a larger accuracy rate can vary; in contrast, a small uncertainty represents a relationship with a certain accuracy.

### 4.3.2. Post-classification Comparison

To analyze the land cover changes between 2007 and 2018, a PCC was performed with the semi-automatic classification plugin on QGIS. The open-source plugin allows two

classified images to be taken as the input (a new map and a reference map), then creating an overlap of these images to cross the data at the pixel level and differentiating the land cover changes according to the differences between the two maps. As the output, a change layer is created, and there is also a table that shows how the pixels move between the classes.

## 5. Results

### 5.1. Comparison of Classifiers

The classification results of the three methods for the two years are presented in Figure 5. The five classes detected in the classification process were cropland; cropland with bare soil; water; vegetated area; and non-vegetation, including urban area, sand, and rocks. Although some differences might exist, generally, the vegetation, non-vegetation, and cropland could be well-identified from different maps, which are globally identical. The vegetation is located in the south and east, with some vegetation near the coastlines, similar to the most important urban areas. In contrast, all of the cropland is in the interior of the peninsula.

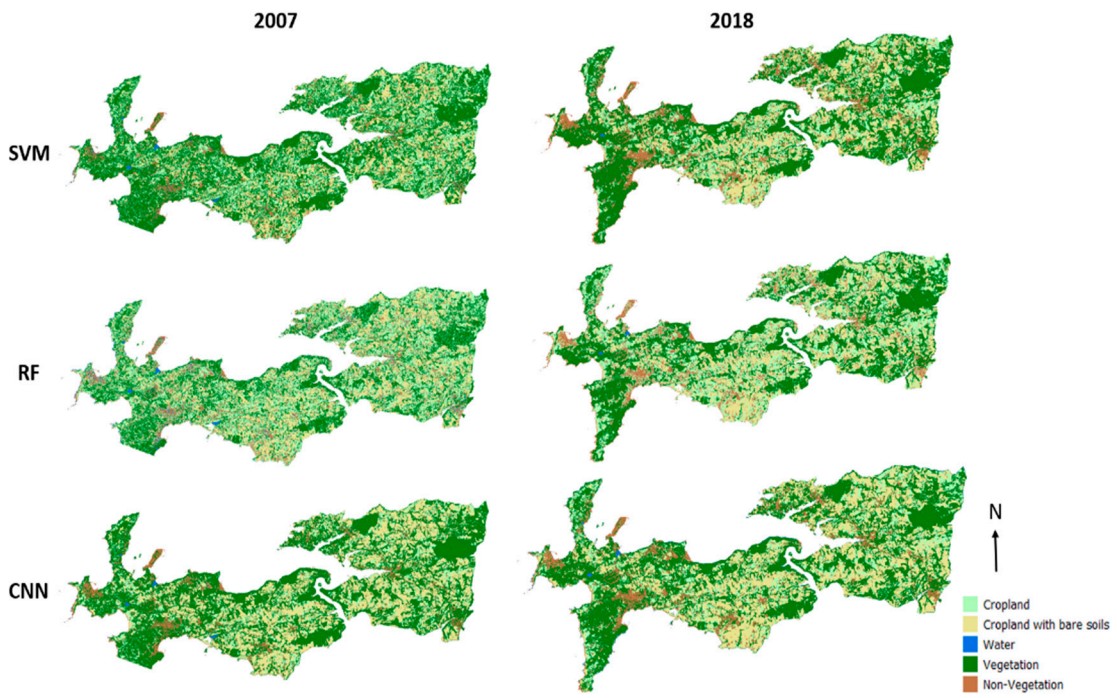

**Figure 5.** Classification results with SVM, RF, and CNN.

To make better comparisons possible, each accuracy assessment in this paper is split into two tables, which are the training accuracy and validation accuracy, allowing for cross-validation to avoid the problem of overfitting or underfitting. The training accuracy was computed and used to improve the model performance and classification quality during the classification processing based on the training dataset; otherwise, the validation accuracy was used with the validation data to evaluate each model's final prediction.

According to Tables 3 and 4, all of the accuracy indices range from 70% to 90%, and the two tables are very similar. Although the training accuracy is slightly higher and more certain (approximately 2–6%), it suggests a good performance and good training of all three models, a strong level of agreement, and a high level of reliability. Beyond that, it is worthwhile to note that the CNN demonstrated better potential (approximately 1–12% higher in accuracy) for the classification of land cover monitoring than RF and SVM in both years, and it is the most stable and certain method, given its low uncertainty of approximately 1.50 for training accuracy and 3.50 for validation accuracy in comparison

with the other methods. In general, the 2007 images have better accuracy indicators and lower uncertainty rates than the 2018 images, and RF achieved better accuracy and lower uncertainty in the 2007 images than in the 2018 (e.g., 80.23 ± 03.87% and 70.51 ± 08.38% for 2007 and 2018, respectively, in the validation accuracy assessment), which is the opposite of SVM (e.g., 77.03 ± 04.36% and 78.14 ± 06.40% for 2007 and 2018, respectively, in the validation accuracy assessment). The most reasonable explanation is that the 2018 images have a rougher resolution than the 2007 images, and so fewer pixels are present in each segment, and since the SVM needs fewer samples and pixels to train the model, it achieved a better performance with the 2018 images.

**Table 3.** Training overall accuracy (%).

| Methods | 2007 | 2018 |
| --- | --- | --- |
| RF | 82.72 ± 01.79 | 76.78 ± 03.40 |
| SVM | 77.17 ± 02.20 | 81.14 ± 03.18 |
| CNN | 89.15 ± 01.36 | 83.16 ± 01.64 |

**Table 4.** Validation overall accuracy (%).

| Methods | 2007 | 2018 |
| --- | --- | --- |
| RF | 80.23 ± 03.87 | 70.51 ± 08.38 |
| SVM | 77.03 ± 04.36 | 78.14 ± 06.40 |
| CNN | 83.11 ± 03.27 | 79.85 ± 03.58 |

The PA and UA of each class with the three methods in both years are listed in Tables 5 and 6. Table 5 presents the satisfactory training accuracy of both models (approximately 70–90%, with a few acceptable exceptions, such as cropland, which has PA and UA values of approximately 40–50%), which indicates that the three models used in the classification: SVM, RF, and CNN, were generally well-trained. Even though the training accuracy and the validation accuracy are constantly approximate, as shown in Tables 3 and 4, the training accuracy is very slightly more accurate and certain than the validation accuracy (approximately 1–10% higher). This suggests a slight overfitting problem in the models. Overfitting, which can be revealed when the training accuracy is significantly greater than the validation accuracy, occurs when the model for the classification is too close and too well-adapted to the training data, in such a way that it is not capable of processing and fitting additional data or making a proper prediction for global images. Nevertheless, the presence of the overfitting problem is not an important obstacle in our study because the differences between the training and validation accuracies are acceptable (between 1% and 10%). Additionally, the validation accuracy always has a higher uncertainty, which indicates that the real accuracy rate may vary to a large extent.

**Table 5.** Training producer's accuracy and user's accuracy by class.

| Land Cover Types | | | 2007 | | | 2018 |
|---|---|---|---|---|---|---|
| Cropland | RF | PA<br>UA | 66.45 ± 08.68<br>40.92 ± 03.90 | RF | PA<br>UA | 78.89 ± 04.48<br>43.71 ± 04.21 |
| | SVM | PA<br>UA | 46.06 ± 06.58<br>51.31 ± 04.79 | SVM | PA<br>UA | 69.91 ± 04.74<br>59.83 ± 05.09 |
| | CNN | PA<br>UA | 79.88 ± 04.53<br>75.25 ± 04.86 | CNN | PA<br>UA | 71.78 ± 04.62<br>71.34 ± 05.06 |
| Cropland (with Bare Soil) | RF | PA<br>UA | 46.34 ± 04.75<br>98.79 ± 01.06 | RF | PA<br>UA | 35.77 ± 05.33<br>91.89 ± 02.65 |
| | SVM | PA<br>UA | 36.15 ± 03.54<br>96.73 ± 01.75 | SVM | PA<br>UA | 52.23 ± 07.21<br>94.78 ± 02.08 |
| | CNN | PA<br>UA | 92.24 ± 02.15<br>92.99 ± 02.09 | CNN | PA<br>UA | 86.54 ± 02.51<br>84.72 ± 02.97 |
| Water area | RF | PA<br>UA | 100<br>41.18 ± 16.54 | RF | PA<br>UA | 100<br>100 |
| | SVM | PA<br>UA | 58.75 ± 48.12<br>72.22 ± 20.69 | SVM | PA<br>UA | 100<br>100 |
| | CNN | PA<br>UA | 100<br>100 | CNN | PA<br>UA | 100<br>71.43 ± 33.47 |
| Vegetation (except crop) | RF | PA<br>UA | 91.50 ± 00.73<br>89.59 ± 02.24 | RF | PA<br>UA | 63.40 ± 06.00<br>88.56 ± 02.22 |
| | SVM | PA<br>UA | 94.93 ± 00.61<br>78.32 ± 02.61 | SVM | PA<br>UA | 65.04 ± 06.45<br>85.26 ± 02.30 |
| | CNN | PA<br>UA | 92.45 ± 01.39<br>90.54 ± 01.90 | CNN | PA<br>UA | 87.58 ± 01.87<br>85.81 ± 02.03 |
| Non-vegetation | RF | PA<br>UA | 81.79 ± 06.72<br>82.14 ± 05.36 | RF | PA<br>UA | 98.22 ± 00.52<br>77.87 ± 05.21 |
| | SVM | PA<br>UA | 38.73 ± 08.16<br>81.71 ± 05.73 | SVM | PA<br>UA | 98.99 ± 00.39<br>80.38 ± 04.83 |
| | CNN | PA<br>UA | 70.71 ± 06.62<br>92.99 ± 04.06 | CNN | PA<br>UA | 68.08 ± 05.64<br>84.29 ± 04.92 |

**Table 6.** Validation producer's accuracy and user's accuracy by class.

| Land Cover Types | | | 2007 | | | 2018 |
|---|---|---|---|---|---|---|
| Cropland | RF | PA<br>UA | 49.63 ± 14.63<br>38.27 ± 07.48 | RF | PA<br>UA | 88.71 ± 06.54<br>54.01 ± 08.35 |
| | SVM | PA<br>UA | 43.21 ± 12.41<br>49.58 ± 08.98 | SVM | PA<br>UA | 63.20 ± 09.69<br>66.67 ± 09.29 |
| | CNN | PA<br>UA | 61.86 ± 09.38<br>65.79 ± 10.67 | CNN | PA<br>UA | 61.34 ± 08.55<br>72.60 ± 10.23 |
| Cropland (with Bare Soil) | RF | PA<br>UA | 47.04 ± 09.65<br>94.68 ± 04.54 | RF | PA<br>UA | 25.69 ± 07.37<br>93.48 ± 05.05 |
| | SVM | PA<br>UA | 35.19 ± 06.81<br>96.39 ± 04.02 | SVM | PA<br>UA | 47.18 ± 08.56<br>98.06 ± 02.66 |
| | CNN | PA<br>UA | 89.96 ± 04.85<br>86.76 ± 05.07 | CNN | PA<br>UA | 90.09 ± 04.81<br>82.55 ± 06.09 |
| Water area | RF | PA<br>UA | 100<br>60.00 ± 30.36 | RF | PA<br>UA | 100<br>50.00 ± 69.30 |
| | SVM | PA<br>UA | 100<br>85.71 ± 25.92 | SVM | PA<br>UA | 100<br>62.07 ± 30.54 |
| | CNN | PA<br>UA | 100<br>100 | CNN | PA<br>UA | 100<br>70 ± 25.05 |
| Vegetation (except crop) | RF | PA<br>UA | 90.66 ± 01.48<br>87.22 ± 04.88 | RF | PA<br>UA | 67.21 ± 13.96<br>91.28 ± 03.96 |
| | SVM | PA<br>UA | 94.91 ± 01.15<br>78.42 ± 05.19 | SVM | PA<br>UA | 65.08 ± 05.30<br>86.94 ± 04.43 |
| | CNN | PA<br>UA | 88.36 ± 03.31<br>85.00 ± 04.52 | CNN | PA<br>UA | 83.79 ± 04.14<br>81.86 ± 05.15 |
| Non-vegetation | RF | PA<br>UA | 84.10 ± 10.16<br>77.78 ± 12.15 | RF | PA<br>UA | 97.87 ± 01.28<br>65.38 ± 12.93 |
| | SVM | PA<br>UA | 45.59 ± 19.51<br>78.05 ± 12.67 | SVM | PA<br>UA | 99.08 ± 01.31<br>74.07 ± 11.69 |
| | CNN | PA<br>UA | 66.06 ± 12.64<br>90.91 ± 09.81 | CNN | PA<br>UA | 63.29 ± 11.88<br>73.68 ± 14.00 |

Among the three methods, the CNN remains the most stable and accurate method, and all of the values range between 70% and 100%. Among the classes, water areas were very well predicted but also very extensive (usually with an accuracy between 50% and 70% with a large uncertainty and 100% ) by means of their distinctive spectral signature. Cropland had training accuracy indicators between 50% and 70%, with CNN having the best

performance in this class (approximately 70–80% in training accuracy and approximately 60–70% in validation accuracy). Of the other two methods, RF was mostly more accurate than SVM (approximately 5–20% higher). In addition, they performed better on the images from 2018 than those from 2007, with a stable uncertainty (approximately 10–20%); the validation accuracies were still approximate and slightly lower. Crops are easily confused with vegetation, which might explain the low accuracy of this classification. Croplands with bare soil are more correctly classified than croplands with plants (with a 20–50% higher rate), and all UA are significantly higher than PA, with a 50% to 70% difference, which suggests that fewer errors of commissions were made during the classification. Except for the CNN, both the UA and PA ranged from 82% to 92%, with less uncertainty at approximately 2.50 in the training accuracy and approximately 5 in the validation accuracy. Even though vegetation has the potential to be confused with crops, it was still the best-predicted class besides water, and the accuracy indicators achieved approximately 80–95%, except for the PA of RF and SVM in 2018, which were approximately 65%. It can be assumed that some errors of omission were made during this classification. The non-vegetation class includes all types of urban land use, sand, and rock; hence, it is globally well-classified due to its particular spectral signature, especially with the 2018 images. The accuracies in general ranged from approximately 70% to 98% in the training accuracy and from approximately 63% to 97% in the validation accuracy. In this class, the fact that the PA is considerably greater than the UA reveals the error of omission, except for the classification of SVM in 2007 and both CNN classifications, which suggests an error of commission instead. In all cases, the CNN was always the most stable and reliable method.

## 5.2. LCLU Detection Changes (2007–2018)

The land use change map resulting from the PCC is shown in Figure 6.

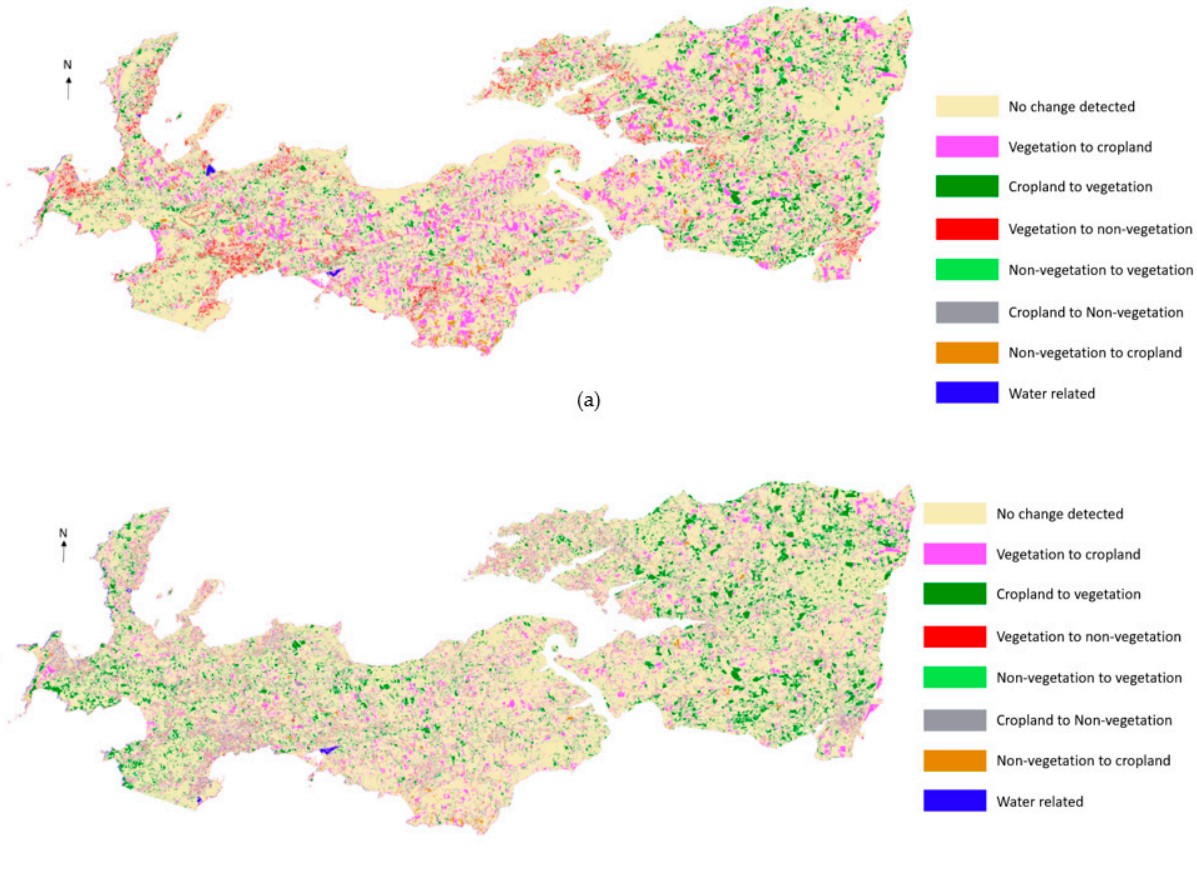

**Figure 6.** *Cont.*

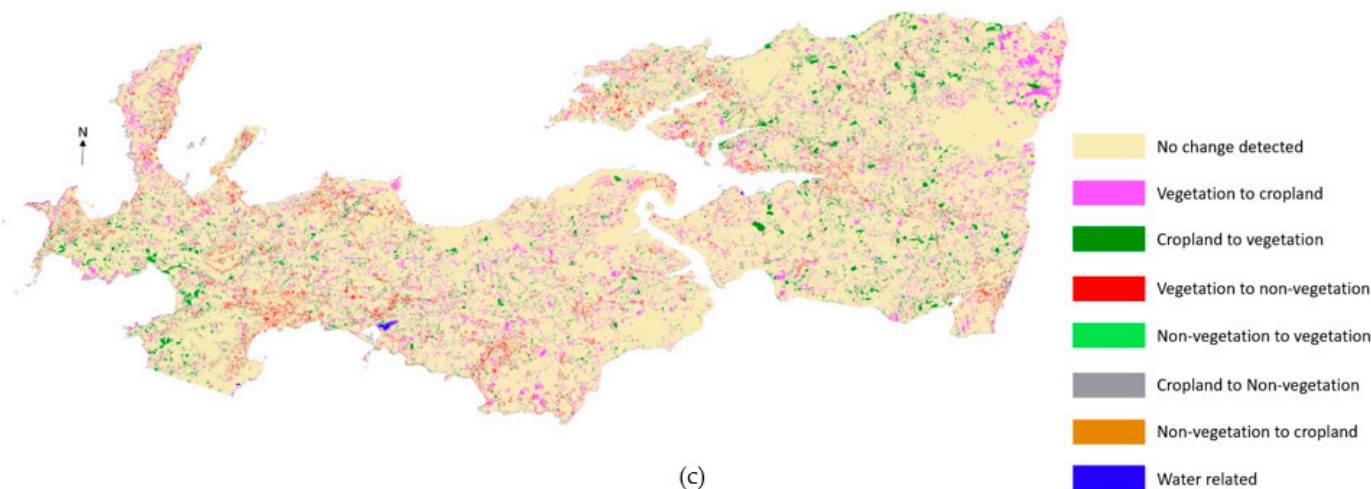

**Figure 6.** Land cover change maps with the classification of SVM (**a**), RF (**b**), and CNN (**c**).

In addition to the problems of the uncertainty and error values caused by classification and spatial resolution differences between two satellite images, multiple changes between the two classes and between the two years (2007 and 2018) were detected. Land cover change maps revealed the status of each pixel, which stayed in the same class, meaning no change, or changes to another class, or another LCLU. Some changes can be seen among the three land cover change maps with different classifications; for example, many croplands were transformed into vegetation, and vegetation was changed to cropland according to the maps of RF. The SVM maps indicate numerous transformations from vegetation to cropland and vegetation to non-vegetation. However, generally, the cropland surface has slightly increased in the peninsula, and many vegetation areas have trun into cropland, according to the land cover change map. At the same time, many cropland areas have become vegetation areas. However, there may be confusion between vegetation and crops due to the different acquired dates of the two images. Therefore, many of the new vegetation areas are most likely growing crops. The third most important land cover change is vegetation to non-vegetation, which mainly took place near urban areas on the coast, especially in the south, where tourism is the most developed. Some details of the three main land cover change types based on the classification results of the CNN, which is the most stable of the three methods, are shown in Figure 7, with comparisons between 2007 and 2018.

Table 7 presents the evolution of the surface of each class between 2007 and 2018 with their proportion in the total surface area of the peninsula, the surface area of each type of land use change and the proportion of each type in the total surface area.

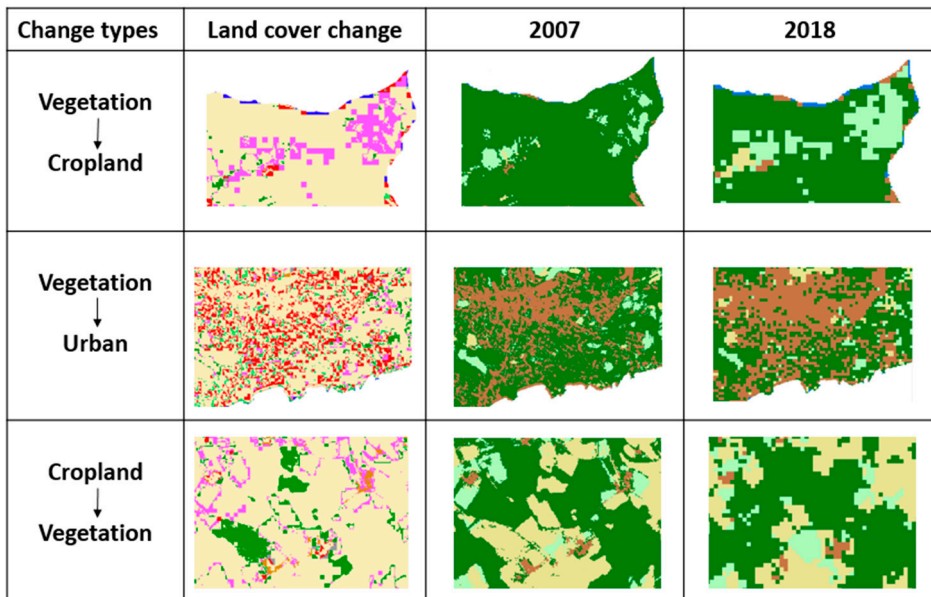

**Figure 7.** Details of the three main land cover change types of the CNN classification with comparisons between 2007 and 2018.

**Table 7.** Land cover change area and proportion of the change type with the three methods of classification.

| Land Cover Change | SVM | | RF | | CNN | |
|---|---|---|---|---|---|---|
| | Area (km$^2$) | Proportion (%) | Area (km$^2$) | Proportion (%) | Area (km$^2$) | Proportion (%) |
| No change | 238.74 | 65.33% | 243.40 | 66.77% | 282.51 | 77.37% |
| Vegetation to Cropland | 45.55 | 12.47% | 31.90 | 8.73% | 35.71 | 9.78% |
| Cropland to vegetation | 43.55 | 12.03% | 49.94 | 13.67% | 23.96 | 6.55% |
| Vegetation to non-vegetation | 20.68 | 5.66% | 5.14 | 1.41% | 12.43 | 3.40% |
| Non-vegetation to vegetation | 1.42 | 0.39% | 3.13 | 0.86% | 3.15 | 0.86% |
| Cropland to non-vegetation | 9.11 | 2.49% | 15.12 | 4.14% | 2.63 | 0.72% |
| Non-vegetation to cropland | 4.61 | 1.26% | 12.17 | 3.33% | 2.63 | 0.72% |
| Water related | 1.34 | 0.37% | 4.04 | 1.11% | 2.14 | 0.59% |
| Total | 365.42 | 100% | 365.42 | 100% | 365.42 | 100% |

In addition to the confusion between growing crops and vegetation, our previous results can be confirmed by Table 7. The table ranges from the most important class with the greatest proportion of land cover change to the least changed class, with the two types of cropland assembled into one class to facilitate the comparisons. The majority of the land in the peninsula retained the same LCLU between 2007 and 2018, and at least 65% to 66% of the area remained unchanged according to the SVM and RF classifications. However, the CNN indicated that approximately 77% of the surface of the Crozon Peninsula did not change between the two years, which is more important. Cropland is clearly increasing: 12.45% of the vegetated area has been converted into cropland, according to SVM classification; however, this transformation is less important according to the RF and CNN classifications, which show approximately 8% and 9%. In contrast, much cropland was identified as vegetation in 2018, more with the SVM and RF classifications (12–13%) than with the CNN (6.55%), which could confuse vegetation and crops due the different acquired dates of the two images. Undoubtedly, non-vegetation, which includes urban areas, has certainly

gained surface area from vegetated areas over the 11-year period by agreement of the three classifications, even though RF presents a lower land cover change value (1.41%) than SVM and CNN (5.66% and 3.40%, respectively). For the RF and SVM classifications, non-vegetation was developed from cropland as well (2.49% and 4.14% in SVM and RF, respectively). A small part of the non-vegetated area was classified as cropland in 2018 in all three models (1.26% in SVM, 3.33% in RF, and 0.72% in CNN); however, it might have been confused with bare soil and non-vegetated areas such as concrete. Finally, the last two classes (non-vegetation to vegetation and all water-related areas) have very low proportions, approximately 0.30–1% in land cover change, which is likely due to the rising tides and increasing water storage in the mid-summer and to planting of small areas, such as in the city.

To conclude, according to the models, the cropland surface has slightly increased, and non-vegetation areas have sharply grown in the 11-year period. The dramatically increasing urbanization of the peninsula, requiring more cropland to address the population growth and tourism development, has resulted in a rapid decrease in vegetation surface area.

## 6. Discussion

### 6.1. LCLU Classification

In this study, three different algorithms were applied to two high spatial resolution satellite images from 2007 and 2018, which were both acquired in the growing season, to map LCLU changes in the Crozon Peninsula, a highly fragmented region. Our objective was to map different LCLUs (cropland, water, vegetation and non-vegetation, including urban land use) and then map and monitor LCLU changes between two years. Another important aspect in the application of the machine learning methods was to recognize the specific type of change when collecting samples for training.

Three classification algorithms (SVM, RF, and CNN) were used, and all of them achieved a good accuracy level, with the overall accuracy ranging from 70% to 90%, despite the complex landscape and small field size. Two machine learning methods, RF and SVM, are object-based approaches, and features other than spectral values play an important role in the classification.

The RF and SVM models both performed well for the LCLU classification; nonetheless, the CNN obviously is better suited to performing classification in our study area, as indicated by the accuracy assessments. According to the results presented in Figure 5 and the statistical evaluations of accuracy provided in Tables 3–6, the proposed method (CNN) generally performs best regardless of the type of dataset and accuracy index. Therefore, the CNN has proven to be a feasible, reliable method with remarkable performance for precisely mapping LCLU and analyzing the changes. Our experiments have shown the superiority of the CNN over other state-of-the-art machine learning classifiers in terms of classification accuracy. However, some important considerations regarding its effectiveness are worth discussing. Previous applications of CNN models have tended to emphasize the complexity of these models compared to RF models and SVMs. In this case, parameter tuning and optimization are often performed by cross-validation for CNN algorithms. However, in some cases, CNN models can have millions of weights to optimize at each iteration [83]. In such situations, training these models can be tedious. Manual tuning or rules of thumb for cross-validation should be implemented in this case. This manual manipulation could have repercussions on the accuracy of the model. A well-known solution is transfer learning [84]. In this case, instead of a model being trained from scratch, pretrained models are retrained on the user's classes of interest. Pretrained models allow for better accuracy [85]. In our study, the deep model was very useful for generalization.

### 6.2. Accuracy Assessment

In accordance with Table 4, the highest OA was obtained by applying the CNN algorithm to 2007 (83.11 ± 03.27). The RF gives the lowest OA, 70.51 ± 08.38. The SVM showed intermediate values between 77.03 ± 04.36 (2007) and 78.14 ± 06.40 (2018).

Regarding the PU and UA (validated results) by class, the best results were obtained with SVM for the PA in 2018 for the non-vegetation class (99.08 ± 01.31; except for the water class). The worst results were always obtained with the SVM for the PA of the cropland class (with bare soil), which was 35.19 ± 06.81.

However, the lower accuracy occurred for 2018, and we deduced that the spatial resolution of the image is a crucial part of classification that can explain the differences between the SVM and RF's overall accuracy in the different years. The RF performed better on the 2007 data with a 2 m spatial resolution SPOT 5 image; in contrast, SVM achieved a better accuracy in 2018 with a 10 m spatial resolution Sentinel 2 image. Among all of the classes, except for the water areas, which have a very different spectral signature than the other classes, vegetation was the best-detected class, most likely because it occurred on the greatest part of the study area; therefore, it also had the largest sample dataset, since all of the samples were randomly and evenly selected in the images. Non-vegetation areas that are mostly urban land, rocks, and sand were relatively simple to discriminate. Cropland with bail soil was better-classified than planted cropland. Misclassification largely occurred between the vegetation and crops due to their spectral signature similarities, especially during the growing season, and they were spatially approximate; some croplands were small and intermixed with trees or shrubs.

The choice of a good classifier is very important, but at the same time, the features extracted from the image are also important. GEOBIA techniques allow the use of hand-created features in the classification phase. The number and choice of features clearly influence the final classification. At the same time, the features of an RF and SVM are learned automatically from the input data during training. The features automatically learned by RF and SVM based on the spectral, contextual and spatial property classes increased the generalization capabilities of the models.

*6.3. LCLU Changes Detection (2007–2018)*

CD techniques can be grouped into two types of objectives: change enhancement and change "from-to" information extraction. In this study, the detection and direction of the changes were processed by applying PCC on a pixel-by-pixel basis through SVM, RF, and CNN classification, with the best performances of the change classes detection between the series of multitemporal images. The multitemporal images were stacked together and then classified directly to detect land cover changes. This work presents a CD protocol that allows reliable PCC to account for the classification accuracies, landscape heterogeneity, and pixel sizes. However, the accuracy of the final change map depends on the quality of each individual classification [86–88]. Errors in the individual maps are additive in the combination (change mapping). In connection with this error question, Liu and Zhou, 2004 [89] proposed a set of rules for the probability of changes from one class to another based on field knowledge. They used these rules to separate "real changes" from possible classification errors. Thus, they determined the accuracy of trajectory changes by arguing the rationality of the changes through a PCC.

Our classification results showed that it is possible to map land use with different algorithms and analyze land use changes between two years. First, increasing the cropland surface indicates that agricultural activities remained an important economic sector in the peninsula, and there were essentially no signs of abandoned agricultural land during the study period. Second, non-vegetation areas increased dramatically due to urbanization, especially some coastal cities that are highly frequented by tourists, since tourism is highly developed in the peninsula. The very dense population corresponds to a high level of artificialization of the territory, which is growing faster than the national average, fueled by a construction of housing and nonresidential premises. This human concentration also implies the progression of urbanization toward the hinterlands, where the construction of housing and the arrival of new residents increased significantly. Artificialization is the main change that has affected the coastal zone of the peninsula, with preferential locations around the major urban centers and on certain coastal sectors. Despite the

regulatory protection established by the Littoral Law, the changes are also important in the 100 m band nearest to the sea and then decrease as one moves away from it. In 1986, the Littoral Law provided an initial regulatory response to the need to control the anarchic development of construction on the coast. One of the most significant consequences of development has been the drastic reduction of the vegetation surface. Vegetation has been removed for two main reasons: increasing agricultural activities and urban land growth. Therefore, economic development can have negative social and economic implications on the peninsula; in addition, environmental conservation and protection are required.

### 7. Conclusions

CD methods involve analyzing the state of a specific geographic area to identify variations from images taken at different times. With satellite remote sensing, high spatial and spectral resolution images are recorded and used to analyze the scales of changes. In this study, in order to detect multiannual change classes between the series of multitemporal images using a pixel-by-pixel PCC technique, three different well-known and frequently used algorithms, including two machine learning algorithms (SVM and RF) and one deep learning algorithm (CNN), were tested on two high spatial resolution satellite images. RF and SVM were applied with an object-based approach, which requires a segmentation step to create subpixel-level objects to avoid the error of mixed pixels since the study area was mainly covered by small fields. The inclusion of the CNN significantly improved the classification performance (5–10% increase in the overall accuracy) compared to the SVM and RF classifiers applied in our study.

Our results showed that the use of remote sensing for complex multiannual small-scale LCLU change studies was completely reliable. The study resulted in two maps that showed five different land uses (cropland, cropland with bare soil, water, vegetation and non-vegetation) in 2007 and 2018 with high accuracy. In particular, the CNN had an overall accuracy that ranged from 80 to 90%, making it the most suitable algorithm in our case, even though RF and SVM also achieved good accuracy levels.

The results may also lead to the conclusion that economic development is rapidly occurring in the peninsula, manifested as urban land and tourism growth, increasing the agricultural activities and grossly decreasing the vegetative areas. Hence, environmental protection measures are demanded for the future. In this context of change, the coastal zones of the peninsula tend to specialize socially and economically, and the maintenance of the agricultural areas, as well as the preservation of the natural areas, are both more sensitive and more complex. Moreover, it appears that the change in land use must be understood in the context of climate change, which is a factor in the aggravation of risks (e.g., flooding and, coastal risks), especially in the sectors that are most subjected to urbanization pressures.

Although we observed relatively high classification accuracies, several uncertainties and limitations persisted. The first is the misclassification between vegetation and planted croplands: the very similar spectral characteristics that they share and their geographical localization lead to this confusion. Second, the two classifications were based on two images with different spatial resolutions; thus, some errors of the land use change analysis could have been induced. Third, useful cloud-free satellite images of the growing season were not easy to obtain in our study area; therefore, a series of annual mappings with more precision was not performed in the study. Hence, some recommendations can be made for further studies, such as applying more vegetation indices or using hyperspectral images to differentiate between vegetation and planted croplands or exploring the potential of synthetic-aperture radar images as a supplement to the traditional optical images on cloudy days.

**Author Contributions:** Conceptualization, S.N. and G.X.; methodology, G.X.; software, G.X.; validation, S.N.; formal analysis, S.N.; investigation, G.X.; resources, G.X.; data curation, G.X.; writing—original draft preparation, G.X.; writing—review and editing, S.N.; visualization, S.N.; supervision,

S.N.; project administration, S.N.; funding acquisition, S.N. All authors have read and agreed to the published version of the manuscript.

**Funding:** This research was funded by Fondation de France.

**Conflicts of Interest:** The authors declare no conflict of interest.

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
