# Peer review of "Mapping and Monitoring of Land Cover/Land Use (LCLU) Changes in the Crozon Peninsula (Brittany, France) from 2007 to 2018 by Machine Learning Algorithms (Support Vector Machine, Random Forest, and Convolutional Neural Network) and by Post-classification Comparison (PCC)"

_remotesensing, doi:10.3390/rs13193899_

Round 1

Reviewer 1 Report

The article has been written in a reliable manner. Title of the article "Mapping and monitoring of land cover / land use (LCLU) 2 changes in the Crozon Peninsula (Britanny, France) from 2007 3 to 2018 by machine learning algorithms (support vector ma-4 chine, random forest, and convolutional neural network ) and 5 by postclassification comparison (PCC) ”generally reflects its content and does not raise any doubts.
The authors exhaustively explained the methods they used and supported them with the relevant literature. Due to new technologies, I consider the literature used in this article to be relevant and comprehensive. The discussion of the results seems to be done and presented correctly. It takes into account various factors causing an increase and a decrease in the accuracy of calculations and the quality of the analyzes performed.
However, it cannot agree with the content of the applications in line 696, in which the paragraph 'The results also indicate that economics develop at a fast rate in the peninsula, which 696 is manifested as urban land and tourism growth, increasing agricultural activities, and 697 grossly decreasing vegetative areas. " The authors may comment on the LCLU, but they should not go any further - pointing out that these are issues of economic development. Photogrammetry cannot tell this, and it is a slight exaggeration on the part of the authors. Of course, such a situation, as described by the authors, probably occurs, but they are not present in the publication. I propose to drop these words.

Author Response

Thank you for your time, we appreciate your suggestion and help. Some problematic parts have been corrected or deleted, and we hope that we can answer your questions with this report.

Reviewer 2 Report

In this paper, the authors have not clarified the contributions of the paper.

What are the two or three research questions that are addressed in this paper?

In the abstract and at later parts of the paper, the authors refer to CNN model but do not clarify whether it is model proposed by the authors.

In line 33, the authors state that "nearly half of the world's population", but do not provide any reference to this information.

In line 85, the authors write "Always in a coastal", it should be corrected.

In line 108, it is unclear what the authors mean by "CNN, a recently developed". As CNN is not recently delveloped.

In line 112, "Niculescu 112 et al., 2018, 2020 [39-40]" is incorrect citation style for this journal.

Nearly all the figures exceed the margin of the template, please correct that.

In line 173, it should be "related" instead of "relative".

In line 182, it is unclear what "other" means here.

In line 212, it is unclear what the authors mean by field surveys.

In line 217, are three satellite images from two dates enough for change detection?

In line 295, what are the basis for selecting these values for the parameters.

Author Response

(The authors gave the same response as above.)

Reviewer 3 Report

This manuscript reports an evaluation of different supervised classification algorithms to detect land cover/land use (LCLU) changes in the Crozon Peninsula between 2007 and 2018, using a post-classification change detection strategy. Although the subject of the manuscript could be interesting, in my view, it does not make any novel contribution and its actual content is far below the publications standards of Remote Sensing.

These are some of the reasons that motivate this unfavorable opinion:  

  • Although, according to the authors, the study area is highly heterogeneous, only five LCLUs are considered to perform the supervised classification of the 2007 and 2018 images: cropland; cropland with bare soil, water area, vegetation and non-vegetation. The manuscript does not describe which crops are included in the cropland and bare-soil cropland classes, nor the type of vegetation included in the vegetation class (if it is tree/arboreal vegetation, shrub, herbaceous vegetation, if it is natural vegetation, etc.). In my opinion, this is a really poor legend, with generalist classes. May be this is why the classification accuracies are quite poor, even for water, with user’s accuracies ranging from 50-70%. These values results worrying taking into account that the spectral behavior of water is completely different from that of other classes.
  • In section 4.2.1.2 the authors highlight the importance of the training sample selection for supervised classification, but do not indicate how many training areas (and their relative surface) have been considered when the SVM and RF classifiers have been applied. However, it is possible to deduce from the manuscript that the training areas selected to classify the multispectral images using the SVM and RF algorithms are different from those selected to classify the images using the CNN classifier. Is it possible to compare the accuracy values obtained using different classifiers if they have not been trained with the same training sample? Moreover, is it possible to compare the accuracy values obtained using different classifiers if the validation sample is not the same?
  • The authors point out in the manuscript that the spatial resolution of the multispectral SPOT-5 images used in this work is 2,5m (lines 195, lines 619-623), which is not exactly true. The 2.5m green, red and near-infrared bands are obtained by merging a panchromatic image acquired at 2.5m resolution with a multispectral image acquired at 10m resolution. The spatial information of this merged image is equivalent to that observed by the panchromatic sensor, but its spectral information is not exactly the same as that the multispectral sensor would observe if it operates at a 2.5m resolution. This is something that the authors should indicate in the manuscript.
  • I do not know the study area, but it is surprising that changes related to water surfaces can affect, as shown in table 6, to more than 200 hectares, in an area of 365km2. Is it not a large extension, especially for a change detection period of 11 years?
  • A detailed explanation about the role of the control parameters of the MRS segmentation algorithm (scale, compactness and shape) is missing in the manuscript. The authors just indicate the values used in this work for these parameters, without any explanation. And the same happens with the control parameters of each classifier.
  • The introduction section does not contain any reference to remote sensing change detection background. I recommend the authors a couple of review papers (Hecheltjen et al, 2014; Afaq and Manocha, 2021). In addition, there are several references that are not appropriates to support the statements which they refer to (line 52, reference [3]; line 54, references [4-6]; line 64, references [9-12]; line 69, reference [1]).
  • The study area section is extremely confusing. The authors provide some geographical data from Crozon Peninsula, but LCLU data from the Department of Finisterre or from the region of Brest (lines 173-180).

References cited in the review:

Hecheltjen, A., Thonfeld, F. and Menz, G., 2014, Recent advances in remote sensing change detection: A review, I. Manakos, M. Braun (Eds.), Land use and land cover mapping in Europe: Practices & Trends, Springer Science, (2014), pp. 145-178.

Afaq, Y. and Manocha, A., 2021, Analysis on change detection technique for remote sensing applications: a review, Ecological Informatics, vol 63, n 101310.

Author Response

(The authors gave the same response as above.)

Round 2

Reviewer 2 Report

Minor language edits are needed, please see line 15, "costal area" should be "coastal area"

Author Response

Hello,

First, thank you so much for your time again, we appreciate your opinion and suggestions. According to your suggestion, we modified/removed a part of our relevant references in the introduction. Secondly, please could you provide more details about the second question "Is the research design appropriate?", so that we can better improve our article.

Kind regards,
Guanyao XIE
